

# Rabbit hindlimb kinematics and ground contact kinetics during the stance phase of gait

Patrick Hall[1], Caleb Stubbs[1], David E. Anderson[2], Cheryl Greenacre[3] and Dustin L. Crouch[1]

[1] Mechanical, Aerospace, and Biomedical Engineering, University of Tennessee - Knoxville, Knoxville, Tennessee, United States
[2] Department of Large Animal Clinical Sciences, University of Tennessee - Knoxville, Knoxville, Tennessee, United States
[3] Department of Small Animal Clinical Sciences, University of Tennessee - Knoxville, Knoxville, Tennessee, United States

Corresponding author
Dustin L. Crouch, dcrouch3@utk.edu

## ABSTRACT

Though the rabbit is a common animal model in musculoskeletal research, there are very limited data reported on healthy rabbit biomechanics. Our objective was to quantify the normative hindlimb biomechanics (kinematics and kinetics) of six New Zealand White rabbits (three male, three female) during the stance phase of gait. We measured biomechanics by synchronously recording sagittal plane motion and ground contact pressure using a video camera and pressure-sensitive mat, respectively. Both foot angle (*i.e.*, angle between foot and ground) and ankle angle curves were unimodal. The maximum ankle dorsiflexion angle was 66.4 ± 13.4° (mean ± standard deviation across rabbits) and occurred at 38% stance, while the maximum ankle plantarflexion angle was 137.2 ± 4.8° at toe-off (neutral ankle angle = 90 degrees). Minimum and maximum foot angles were 17.2 ± 6.3° at 10% stance and 123.3 ± 3.6° at toe-off, respectively. The maximum peak plantar pressure and plantar contact area were 21.7 ± 4.6% BW/cm$^2$ and 7.4 ± 0.8 cm$^2$ respectively. The maximum net vertical ground reaction force and vertical impulse, averaged across rabbits, were 44.0 ± 10.6% BW and 10.9 ± 3.7% BW·s, respectively. Stance duration (0.40 ± 0.15 s) was statistically significantly correlated ($p < 0.05$) with vertical impulse (Spearman's $\rho = 0.76$), minimum foot angle ($\rho = -0.58$), plantar contact length ($\rho = 0.52$), maximum foot angle ($\rho = 0.41$), and minimum foot angle ($\rho = -0.30$). Our study confirmed that rabbits exhibit a digitigrade gait pattern during locomotion. Future studies can reference our data to quantify the extent to which clinical interventions affect rabbit biomechanics.

## INTRODUCTION

The rabbit is a common animal model in musculoskeletal research to, for example, test new potential clinical interventions or study the response of tissues to mechanical stimuli. Some interventions would be expected to affect the motor function of the hindlimb ankle and foot. Examples of such interventions include tenotomy of ankle dorsiflexor (*Abrams*

*et al., 2000*) or plantarflexor (*Nagasawa et al., 2008*; *Reddy, Stehno-Bittel & Enwemeka, 1998*) muscles, immobilization of the knee and/or ankle joint (*Gossman et al., 1986*; *Ponten & Friden, 2008*; *Sjostrom, Wahlby & Fugl-Meyer, 1979*), and release of tendon retinacula to manipulate muscle-tendon moment arms (*Koh & Herzog, 1998b*; *Reddy & Gupta, 2007*). Recently, we adopted a rabbit model to test the feasibility of a new type of ankle-foot prosthesis (*Hall et al., 2021*). For potential clinical interventions, quantifying their effect on motor function will be valuable, if not essential, for achieving clinical translation.

One way to quantify motor function is by measuring biomechanical variables. The effect of an intervention could be determined by comparing biomechanical data between animals that did and did not receive the intervention. Unfortunately, the hindlimb biomechanics of healthy rabbits has not been well characterized. We are aware of only one previous study in which multi-sample data were collected, and those were limited to knee and ankle kinetics (*i.e.*, joint moments) during gait (*Gushue, Houck & Lerner, 2005*). A more comprehensive dataset on normative rabbit hindlimb biomechanics in the literature would (1) make it easier for researchers to determine the effects of experimental interventions on motor function, (2) reduce the number of healthy animals used as experimental controls, and (3) increase basic understanding about animal locomotion.

Two common, broad measures of hindlimb biomechanics that have not been reported for a multi-sample rabbit cohort are joint kinematics and ground contact kinetics. For example, rabbits are considered to have a "plantigrade type" foot morphology (*Kimura, 1996*) but a digitigrade gait pattern during locomotion (*Blair, 2013*; *Volait-Rosset et al., 2019*). Unfortunately, there are no published kinematic data to quantitatively verify such statements about the hindlimb gait patterns of rabbits. In terms of ground contact kinetics, one previous study reported that the net vertical ground reaction force curve during hindlimb stance phase displayed a slight bimodal pattern with a maximum force of nearly 60% body weight (*Gushue, Houck & Lerner, 2005*); However, these kinetic data were presented for only one rabbit and one trial; therefore, it is unclear if the patterns and values are consistent across multiple rabbits and trials.

The goal of our study was to quantify hindlimb ankle and foot kinematics and ground contact kinetics during the stance phase of gait in healthy rabbits (*i.e.*, rabbits that have not received an experimental intervention). Summary results are presented below, and the raw data are available as Supplemental Data. The new biomechanics data will serve as a valuable reference for interventional studies involving rabbits and improve our basic understanding about rabbit locomotion.

## MATERIALS AND METHODS

All animal procedures were approved by the University of Tennessee, Knoxville Institutional Animal Care and Use Committee (protocol #2637). We used a cross-sectional study design, measuring biomechanics in one test session from both hindlimbs of six (three male, three female) healthy, standard laboratory New Zealand White (NZW) rabbits (Charles River Laboratories, Wilmington, MA, USA). The rabbits represented a convenience sample as they were part of another larger, iterative, non-hypothesis-driven study to test a novel orthopedic implant (*Hall et al., 2021*); thus, no power analysis was

performed for either study. Since this was not a clinical trial, there were no separate experimental groups. At the time of testing, the rabbits were 16 weeks old and weighed 2.7 ± 0.33 kg. Each rabbit was considered one experimental unit. Rabbits were housed individually in adjacent crates, fed *ad libitum* with a standard laboratory diet and Timothy hay, and given daily enrichment and positive human interaction.

The setup for our locomotor measurement system included several components (Fig. 1). A four-tile pressure mat (Very High Resolution Walkway 4; Tekscan, Inc., South Boston, MA, USA) was used to record pressure data at 60 Hz. We taped 320-grit sandpaper to the top of the smooth pressure-sensing area to prevent the rabbits from slipping. Each tile was calibrated independently using a 3-kg mass with three separate felt pads that each approximated the geometry and texture of the rabbit hindlimb plantar surface. The mat was placed inside a clear acrylic tunnel, which constrained the rabbits to ambulate along the mat. The mat width (11.2 cm) permitted only unilateral pressure measurements; therefore, to record data for both hindlimbs, we laterally offset the mat in the tunnel and had the rabbit ambulate through the tunnel in both directions, as described below. A camera (1080P HD Webcam; SVPRO, Saarwellingen, Germany), placed three feet away from the clear acrylic panel, captured video at 60 Hz. Video and pressure mat data were synchronized using the Tekscan Walkway software (Tekscan, South Boston, MA, USA).

During a 2-week acclimation period prior to testing, rabbits were trained to ambulate through the acrylic tunnel when given negative reinforcement (*i.e.*, prodding). Then, at the beginning of the test session, we shaved both hindlimbs and marked the metatarsophalangeal (MTP), ankle, and knee joint centers on the lateral side of each hindlimb with black ink (Fig. 2A) to facilitate calculation of joint kinematics from the videos. After marking, we placed the rabbit into a pen with the acrylic tunnel and pressure mat. Each trial began when the rabbit entered the tunnel. A trial was deemed successful if (1) the rabbit continued moving through the entire length of the tunnel without stopping, (2) all hindlimb markers were visible in the video frame during a stance phase, and (3) the entire plantar contact area was within the sensing area of the pressure mat. These criteria were checked both during and after the test session. The session continued until we collected data for 5–7 successful trials of gait through the tunnel in each direction. The direction of gait is a potential confounder, which we address as a limitation in the Discussion section.

All confirmed successful trials—six to 10 trials per rabbit, 49 trials in all were selected for analysis. We calculated joint angles from the video frames corresponding to stance phase using a custom script in MATLAB (Mathworks, Natick, MA, USA). The program used a bottom-hat morphology filter (*imbothat* function in MATLAB) to distinguish the black ink marks and calculate the centroid of each marker position (Fig. 2B). Then, frame-by-frame, we visually verified the marker centroids and, if the centroid location appeared inaccurate, corrected the location by manually approximating the centroid from the still frame. Calculating marker positions from a two-dimensional video frame assumes that all markers lie in a plane that is parallel to the video frame; this was a reasonable assumption given the arrangement of the camera with respect to the acrylic tunnel. Finally, we calculated the angle between the foot segment and ground (*i.e.*, foot angle) and between

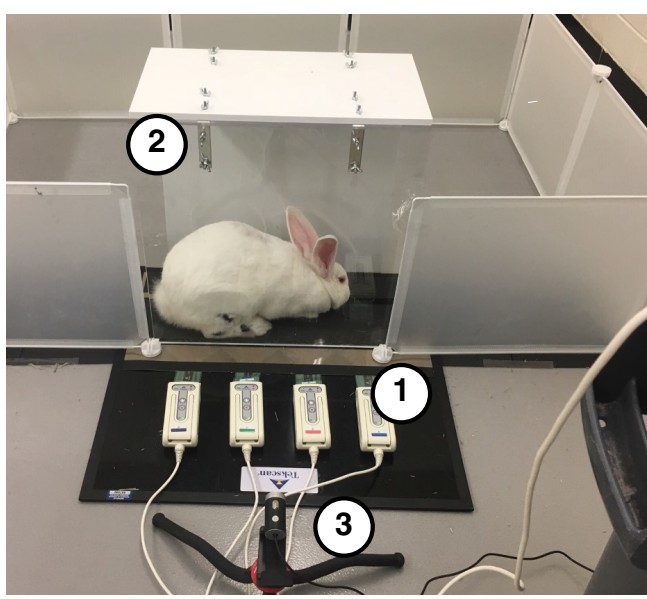

**Figure 1 Testing setup used to collect biomechanical data.** (1) Tekscan Very HR Walkway four for measuring pressure data. (2) Acrylic tunnel for guiding rabbits across pressure mat. (3) A total of 60 Hz Camera for recording sagittal plane kinematics.

the foot and shank segments (*i.e.*, ankle angle) throughout stance (Fig. 2C). Since the camera was approximately level with the ground, we defined the ground as a horizontal line in the video frame. Ankle angles >90° and <90° corresponded to plantarflexion and dorsiflexion, respectively.

Ground contact pressure data were processed using the Tekscan Walkway software. Each hind foot was isolated by drawing a strike box around the corresponding pressure map. From the pressure map in the strike box and for each time point, we computed (1) peak and average plantar pressures, (2) plantar contact area and length, (3) net vertical ground reaction force (vGRF), and (4) vertical impulse. The contact area was calculated at each timepoint as the total geometric area within the strike box for which the pressure was greater than zero. The plantar contact length was the total cranial-caudal length of the plantar surface that contacted the ground at any time during stance; plantar contact length was expressed as a percentage of the total length of the plantar surface, measured from unsuccessful trials during which the rabbit stopped and placed the entire plantar surface on the pressure mat. The vGRF was calculated as the product of foot contact area and the average pressure across the contact area. Vertical impulse was calculated as the time integral of the vGRF curve. Pressure, vGRF, and vertical impulse values were normalized by body weight.

We computed the stance duration for each limb as the length of time for which pressure was greater than zero within the limb's strike box. The timepoints of time-series biomechanical data were normalized by the stance duration; at each normalized timepoint, data were averaged across trials for each rabbit, then averaged across rabbits. Similarly, plantar contact length and vertical impulse were averaged across trials for each rabbit, then averaged across rabbits.

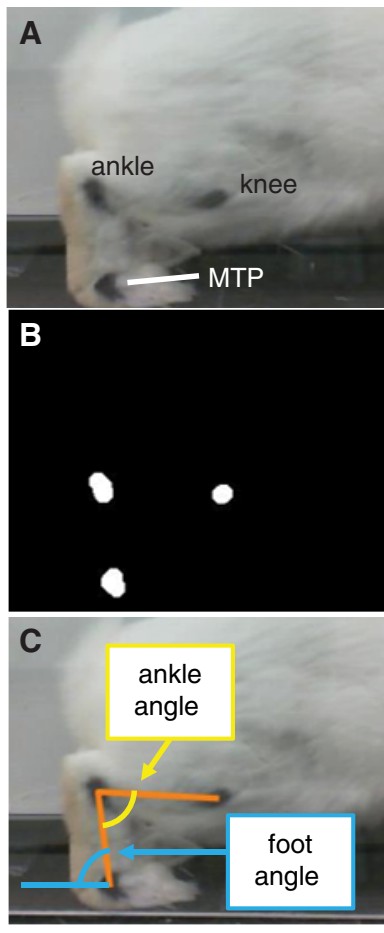

**Figure 2 Motion capture analysis from sagittal plane videos for calculating hindlimb kinematics.**
(A) Ink markers were placed on the lateral aspect of the hindlimb approximately over the knee, ankle, and metatarsophalangeal (MTP) joint centers. (B) A frame-by-frame bottom-hat morphology permitted detection of joint markers. (C) Vectors representing limb segments (orange lines) were drawn between the centroids of the joint markers. Using the dot product and the segment vectors, we calculated the ankle angle (yellow) and foot angle (blue).

We observed that stance duration across trials ($n$ = 49) had a relatively high variation and exhibited a right-skewed distribution (mean = 0.40 s, median = 0.35 s, standard deviation = 0.15 s). Since gait biomechanics are known to vary with gait speed during bipedal (*Lelas et al., 2003*) and quadrupedal (*Boakye et al., 2020*) locomotion, we tested whether stance duration was significantly correlated with select biomechanical outcome variables (Table 1). Given that stance duration was not normally distributed, we used the non-parametric Spearman's rank-order correlation test. We computed Spearman's correlation coefficient ($\rho$) to determine the strength and direction of correlation. Strength was interpreted based on the magnitude of $\rho$ as previously reported (*Chan, 2003*). The correlation was considered statistically significant for $p < 0.05$. For the correlation test, we considered each trial as an independent sample.
**Table 1 Select biomechanical variables and their correlation with stance duration.**

| Biomechanical variable | Mean ± SD (*n* = 6 rabbits) | Correlation with stance duration (*n* = 49 trials) | |
| --- | --- | --- | --- |
| | | Spearman's rho ($\rho$) | *p* |
| Stance duration (s) | 0.40 ± 0.15 | — | — |
| Maximum foot angle (deg) | 123.3 ± 3.6 | 0.41 | 0.003 |
| Minimum foot angle (deg) | 17.2 ± 6.3 | −0.58 | <0.001 |
| Maximum ankle angle (deg) | 137.2 ± 4.8 | −0.13 | 0.363 |
| Minimum ankle angle (deg) | 66.4 ± 13.4 | −0.30 | 0.034 |
| Maximum plantar contact area ($cm^2$) | 7.4 ± 0.8 | 0.03 | 0.815 |
| Plantar contact length (%) | 70 ± 11 | 0.52 | <0.001 |
| Maximum peak plantar pressure (%BW/$cm^2$) | 0.217 ± 0.046 | −0.24 | 0.092 |
| Maximum vGRF (%BW) | 44.0 ± 10.6 | −0.35 | 0.014 |
| Vertical impulse (%BW·s) | 10.9 ± 3.7 | 0.76 | <0.001 |

Note:
Spearman's rank-order correlation test was performed to compute the correlation coefficient, rho ($\rho$). *P*-values were computed to determine the statistical significance of the correlation.

## RESULTS

At foot strike, the ankle angle became increasingly dorsiflexed for the first 38% of stance, then increasingly plantarflexed for the remaining 62% of stance (Fig. 3A); these phases are sometimes referred to as "loading response" and "forward propulsion", respectively (*Li & Hsiao-Wecksler, 2013*). The maximum ankle dorsiflexion angle during stance was 66.4 ± 13.4° (mean ± standard deviation). The ankle was plantarflexed (>90°) at both foot strike (103.1 ± 13.0°) and toe-off (137.2 ± 4.8°), with the maximum ankle plantarflexion angle during stance occurring at toe-off. The foot maintained a positive angle throughout stance, starting at 21.1 ± 6.9°, then decreasing to a minimum foot ankle of 17.2 ± 6.3° at 10% stance (Fig. 3B). Thereafter, the foot ankle gradually increased up to 123.3 ± 3.6° at toe-off, indicating that the foot had rotated beyond a vertical orientation (90°).

Peak plantar pressure over the stance phase was unimodal, reaching a maximum of 21.7 ± 4.6% BW/$cm^2$ at 33% stance (Fig. 4A). By comparison, average plantar pressure over the entire plantar contact area was nearly uniform between 5% and 95% stance, with the mean ranging from 4.3–6.4% BW/$cm^2$ over that interval (Fig. 4B). The maximum peak plantar pressure (21.7% BW/$cm^2$) was over 3× larger than the maximum average plantar pressure (6.4% BW/$cm^2$). The vGRF curve over stance was unimodal, with a maximum of 44.0 ± 10.6% BW that occurred at 27% stance (Fig. 5). Vertical impulse was 10.9 ± 3.7% BW·s.

The plantar contact area curve was unimodal (Fig. 6); the maximum plantar contact area was 7.4 ± 0.8 $cm^2$ and occurred at 21% stance. The plantar contact length was, on average, 70 ± 11% of the total plantar surface length and ranged from 41–95% across all trials. The plantar contact length was shifted entirely toward the cranial aspect of the plantar surface (*i.e.*, toes), indicating that the rabbits exhibited a digitigrade gait pattern.

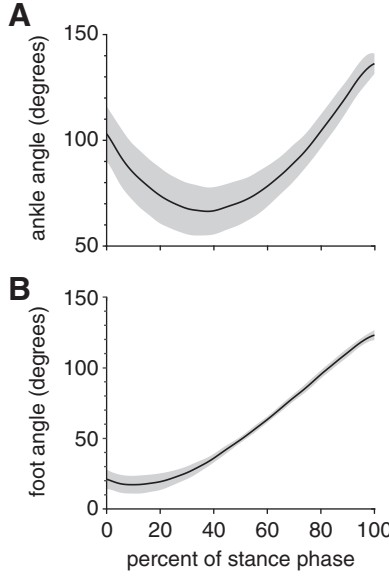

**Figure 3 Joint kinematics of the rabbit hindlimb during stance phase.** Mean (black line) and ±1 standard deviation (grey shaded region) of (A) ankle angle and (B) foot angle. Consistent with previous descriptions of gait (*Li & Hsiao-Wecksler, 2013*), we divided rabbit stance into "loading response" and "propulsion" sub-phases at the maximum ankle dorsiflexion (*i.e.*, smallest value) angle (grey dotted line).

Stance duration was significantly correlated with several biomechanical outcome variables (Table 1). There was a strong, positive correlation between stance duration and vertical impulse ($\rho = 0.76$, $p < 0.001$). Stance duration had fair-to-moderate correlation with minimum foot angle ($\rho = -0.58$, $p < 0.001$) and plantar contact length ($\rho = 0.52$, $p < 0.001$). Maximum foot angle ($\rho = 0.41$, $p = 0.003$), minimum ankle angle ($\rho = -0.30$, $p = 0.034$), and maximum vGRF ($\rho = -0.35$, $p = 0.014$) had fair correlation with stance duration.

## DISCUSSION

Rabbits are among the few mammals that almost exclusively exhibit a bounding or half-bounding gait in which the hindlimbs push off and move together (*Dagg, 1973*). Other mammals, such as rats, mice, cats, and squirrels, use a bounding gait at fast gait speeds but a different gait pattern (*e.g.*, walking) at slow gait speeds. To our knowledge, there are few sources that present biomechanical data for rabbits that are similar to our data; thus, below, we primarily discuss our results in the context of those reported for other quadrupedal species. Though we did not directly measure gait speed, the stance duration of 0.40 ± 0.15 s for our rabbits is consistent with a walking gait speed (about 0.6 m/s) in cats (*Goslow, Reinking & Stuart, 1973*; *Verdugo et al., 2013*).

vGRF data have been reported for only one rabbit (*Gushue, Houck & Lerner, 2005*) and, unsurprisingly, are not representative of the average vGRF values we observed across multiple rabbits and trials. For example, the reported vGRF was bimodal and had a maximum of greater than 50% BW (*Gushue, Houck & Lerner, 2005*); by comparison, the average vGRF curve in our study was unimodal and had a maximum of 44% BW. This

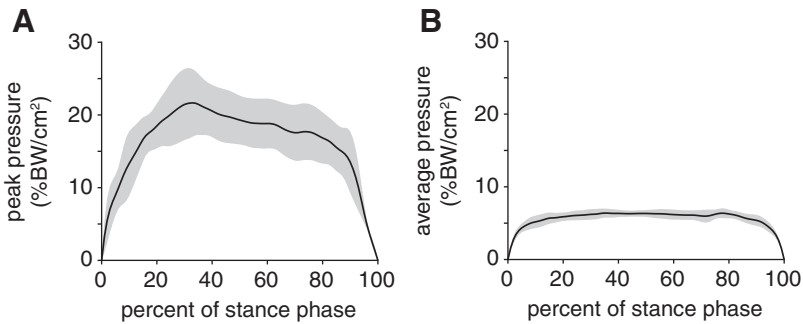

**Figure 4 Plantar pressures of the rabbit hindlimb during stance phase.** (A) Mean (black line) and ±1 standard deviation (grey shaded region) of (B) peak plantar pressure and (C) average plantar pressure. Both pressures were normalized by body weight.

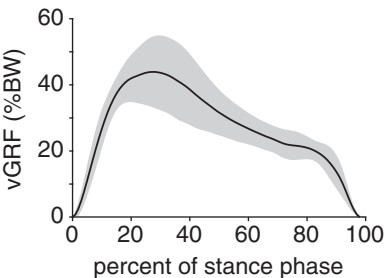

**Figure 5 Net vertical ground reaction force (vGRF) of the rabbit hindlimb during stance phase.** vGRF was expressed as a percentage of body weight (%BW). Black line and grey shaded region indicate the mean and ±1 standard deviation, respectively.

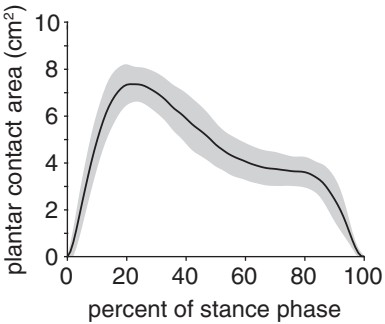

**Figure 6 Plantar contact area of the rabbit hindlimb during stance phase.** Black line and grey shaded region indicate the mean and ±1 standard deviation, respectively.

maximum vGRF was similar to that of cats (*Verdugo et al., 2013*) and dogs (*Besancon et al., 2003*) during walking. Conversely, vertical impulse was lower in the rabbits (~11% BW·s) than in cats (~13% BW·s) (*Verdugo et al., 2013*) and dogs (~18% BW·s) (*Besancon et al., 2003*) during walking.

Compared to maximum ground contact kinetic data, time-series kinetic data have not been widely reported in animals. Studies have reported time-series pressures under different plantar regions in, for example, dogs (*Marghitu et al., 2003*) and bonobos

(*Vereecke et al., 2003*). The time-series pressure curves in dogs were unimodal (*Marghitu et al., 2003*), similar to our rabbits. Time-series ground contact kinetics should be reported more frequently, at least as Supplemental Data, as they provide more detailed information than maximum values.

The plantar contact length data confirmed that rabbits exhibit a digitigrade gait pattern, consistent with previous qualitative descriptions of rabbit gait (*Blair, 2013*; *Volait-Rosset et al., 2019*). Most mammals use a digitigrade gait pattern during locomotion, which is more efficient than the plantigrade gait of humans (*Cunningham et al., 2010*). The rabbits' foot and ankle kinematics were generally similar to those of rats during walking (*Bauman & Chang, 2010*; *Varejao et al., 2002*). Specifically, in both rabbits and rats, the ankle joint angle curve during stance was nearly unimodal. The ankle progressed from an initial, slightly plantarflexed posture to a dorsiflexed posture that peaked around mid-stance, then returned to a plantarflexed posture at terminal stance (*Bauman & Chang, 2010*; *Varejao et al., 2002*). Foot angle is not commonly reported but, qualitatively, rats progressed from a low to high foot angle during stance similar to the rabbits in our study (*Varejao et al., 2002*).

Based on the correlation coefficients (Table 1), there were several statistically significant relationships between stance duration and other biomechanical outcome variables. Specifically, for shorter stance durations, the hindlimb exhibited (1) shorter plantar contact lengths (*i.e.*, more extreme digitigrade gait), (2) a more-vertical foot posture, (3) lesser maximum ankle dorsiflexion, (4) greater maximum vertical force (vGRF), and (5) lesser vertical impulse. In both quadrupeds (*Goslow, Reinking & Stuart, 1973*; *Vilensky, 1987*) and humans (*Fukuchi, Fukuchi & Duarte, 2019*), stance duration decreases as gait speed increases. Assuming that the same was true for the rabbits in our study (though we did not measure gait speed directly), our correlations were generally consistent with those reported previously. For example, the ankle is also less dorsiflexed (*i.e.*, more plantarflexed) at faster walking speeds (*i.e.*, shorter stance durations) in humans (*Fukuchi, Fukuchi & Duarte, 2019*), cats (*Goslow, Reinking & Stuart, 1973*), and pigs (*Mirkiani et al., 2022*). Cats also have greater maximum vertical forces and lesser vertical impulse at faster walking speeds (*i.e.*, shorter stance durations) (*Liu et al., 2020*).

We chose an approach to measure rabbit hindlimb biomechanics that is practical for longitudinal interventional studies in rabbits. For example, we used a video-based motion capture method to quantify hindlimb kinematics from ink-based skin markers. Conversely, state-of-the-art methods track flat or spherical reflective markers that are placed on the skin using adhesive tape. Ink-based markers stay on the skin more reliably than adhesive markers that can fall off during test sessions. However, any markers placed on the skin are prone to skin motion artifact and inaccurate marker placement relative to the joint center. In our study, skin motion and marker placement error contributed to high variation (maximum-minimum) in the lengths of foot (20%) and shank (40%) segments computed from marker locations in each trial. This variation is consistent with the large errors in joint angles computed from skin markers in rats (*Bauman & Chang, 2010*). These errors motivate the use of more reliable motion capture methods, such as reflective markers attached to transdermal bone pins (*Gushue, Houck & Lerner, 2005*) or

radiography (*Bauman & Chang, 2010*). Though reliable, these methods have practical limitations. For example, bone pins are relatively difficult to implement, especially for longitudinal studies, and may interfere with movement either mechanically or by causing discomfort. Bi-plane fluoroscopy can acquire detailed kinematic data (*Koh & Herzog, 1998a*; *Tinga et al., 2018*) but poses a radiation safety risk to the animals and researchers. Finally, both infrared- and fluoroscopy-based methods require relatively expensive equipment that must remain stationary during testing and, thus, are not as accessible, portable, or mobile as video-based methods.

Pressure mats are, in some ways, more convenient to use with quadrupedal animals than force plates, a common alternative. This is because force plates measure the resultant force, which requires that only one limb contacts the plate at a time to distinguish forces among limbs (*Gushue, Houck & Lerner, 2005*; *Jarrell et al., 2018*). Conversely, pressure mats can distinguish among individual limbs even if multiple limbs contact the mat simultaneously (*Sheldon et al., 2019*; *Steiner et al., 2019*). Considering also that force plates are generally smaller than pressure mats, more gait trials may be needed to collect desired ground contact kinetic data with force plates than with pressure mats. However, force plates have notable advantages. For example, force plates measure forces along 6 degrees of freedom and, thus, capture all the directional components of the ground-contact force, such as propulsive forces that occur in the horizontal direction; conversely, pressure mats can only measure the vertical force component. While both pressure mats and force plates provide reliable estimates of variables related to bilateral symmetry and duration, force plates measure absolute force variables, such as maximum vGRF and vertical impulse, more accurately (*Oosterlinck et al., 2010*).

Our study was limited in several ways. First, because our pressure mat was relatively narrow (11.2 cm), we only measured kinematic and pressure data for one hindlimb at a time. In future studies we plan to use a wider pressure mat and two cameras (one on each side) to capture biomechanics data from both hindlimbs simultaneously; this data will allow us to quantify temporal relationships between sides. Second, the sample frequency (60 Hz) of our motion capture setup allowed us to capture stance phase only; in future studies we will use cameras with a higher sample frequency to also capture the high-frequency kinematics of the swing phase. Third, we computed kinematics in the sagittal plane from a single video; this had practical advantages compared to multi-camera systems, as described above, but cannot reveal potential effects of interventions on out-of-plane motion. Fourth, we used a custom MATLAB script to compute kinematics from videos of rabbits with ink-based markers, though more advanced machine-learning-based open-source software, such as DeepLabCut (*Mathis et al., 2018a*; *Mathis et al., 2018b*), are available. Finally, in our correlation analysis, we used stance duration as a surrogate for gait speed; future studies should measure and use gait speed directly since it is more customary in such analyses (*Boakye et al., 2020*; *Lelas et al., 2003*).

## CONCLUSIONS

In conclusion, we have reported new biomechanical data based on hindlimb kinematics and ground contact kinetics from healthy New Zealand White rabbits. Our results showed

that rabbits exhibit a digitigrade gait pattern as described in previous literature. Several biomechanical values and their correlation to stance duration were similar to those observed for other small mammals, such as rats, cats, and dogs. Our results add substantially to the limited existing data on rabbit hindlimb biomechanics. The data can be used as an experimental control to quantify the effect of experimental interventions while reducing the number of animals used for research. Importantly, the data can inform pre-clinical trials of musculoskeletal and other clinical interventions, commonly tested in rabbits, to improve functional outcomes for human and animal patients.

## ACKNOWLEDGEMENTS

The authors thank Dr. Bryce Burton, Dr. Kelsey Finnie, Dr. Lori Cole, and Chris Carter for veterinary care provided for the rabbits in this study, and the Office of Laboratory Animal Care and Animal Housing Facility staff at the University of Tennessee, Knoxville for animal care assistance. Thanks to Dr. Katrina Easton for providing constructive feedback on the manuscript.

### Funding

Research reported in this publication was funded by (1) the Eunice Kennedy Shiver National Institute of Child Health & Human Development of the National Institutes of Health under Award Number K12HD073945, (2) NSF CAREER Award #1944001, (3) a seed grant from the University of Tennessee Office of Research and Engagement, and (4) the University of Tennessee Department of Mechanical, Aerospace and Biomedical Engineering start-up funds. The funders had no role in study design, data collection and analysis, decision to publish, or preparation of the manuscript.

### Grant Disclosures

The following grant information was disclosed by the authors:
Eunice Kennedy Shiver National Institute of Child Health & Human Development of the National Institutes of Health: K12HD073945.
NSF CAREER Award: #1944001.
University of Tennessee Office of Research and Engagement.
University of Tennessee Department of Mechanical, Aerospace and Biomedical Engineering.

### Competing Interests

The authors declare that they have no competing interests.

### Author Contributions

- Patrick Hall conceived and designed the experiments, performed the experiments, analyzed the data, prepared figures and/or tables, authored or reviewed drafts of the article, and approved the final draft.

- Caleb Stubbs performed the experiments, analyzed the data, prepared figures and/or tables, authored or reviewed drafts of the article, and approved the final draft.
- David E. Anderson conceived and designed the experiments, authored or reviewed drafts of the article, and approved the final draft.
- Cheryl Greenacre conceived and designed the experiments, authored or reviewed drafts of the article, and approved the final draft.
- Dustin L. Crouch conceived and designed the experiments, prepared figures and/or tables, authored or reviewed drafts of the article, and approved the final draft.

### Animal Ethics

The following information was supplied relating to ethical approvals (*i.e.*, approving body and any reference numbers):

IACUC at the University of Tennessee, Knoxville provided full approval for the research.

### Data Availability

The raw and processed data are available in the Supplemental File.

### Supplemental Information

Supplemental information for this article can be found online at http://dx.doi.org/10.7717/peerj.13611#supplemental-information.

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
