# Peer review of "Rabbit hindlimb kinematics and ground contact kinetics during the stance phase of gait"

_PeerJ, doi:10.7717/peerj.13611_

## Round 0.1 · original submission · Major Revisions

Dear Dr Hall,

Thank for your submission, “Rabbit hindlimb kinematics and ground contact pressure during the stance phase of hopping gait”. We’ve received reports from two reviewers, both of whom agree that some revisions are required.

All reviewers primarily raise concerns about the research methodology, noting several cases in which more details are required about the data collection protocol and the possible need to include measures of locomotor speed as a covariate. Reviewer 1 also notes that more context is needed in the introduction and discussion regarding the proposed similarities between human and rabbit gait. Please take a careful look at all comments and add clarifications, caveats, or additional analyses where required.

I had some additional concerns not mentioned by the reviewers, referenced by author-supplied line number:

(Lines 104-105) What was the length of the tunnel? How did this compare to average rabbit stride lengths?

(Line 124) The authors mention using a “top-hat” morphology filter, but the figure legend mentions a “bottom-hat” filter. Please clarify.

(Line 137-140) I’m a little unclear on why the authors chose to calculate knee joint position in this way. I understand the distinction the authors are making between joint angles and limb joint center positions, but I think for that distinction to really hold sway it’s important to define the control parameter for the animal. In other words, which is more important for the locomotor task at hand – achieving a given angular position or a given linear position? Both can be valid measures of locomotor kinematics in a control framework. Additionally, unless I’m mistaken, the method used to calculate knee joint position seems a little flawed, in that calculating ankle angle requires the use of knee joint position and then ankle angle is itself used to back-calculate knee joint position. Isn’t this tautologous? Or is the original ankle angle based on the skin-markered knee joint position?

(Line 188) Is it possible to compare the maximum length of the pressure area to the morphological length of the rabbits’ feet? This would give a number that could establish the degree to which rabbits are truly plantigrade (e.g., “At the beginning of stance phase, rabbits contacted the ground with X% of foot length”).

(Lines 244-248) 3D kinematics are not possible without more cameras. This is true whether you’re using DeepLabCut or any other software method to calculate joint kinematics.

(Figures 1 and 6) Please use letters to delineate different panels within the same figure.

I look forward to receiving your revised manuscript.

Jesse W. Young
Department of Anatomy and Neurobiology
Northeast Ohio Medical University

Reviewer 1 ·

Basic reporting

The authors use clear, professional language, and have written an appropriate article in style and format, that is self-contained with relevant results and hypotheses.

Experimental design

This experiment is appropriate for PeerJ in scope and aims and fills a gap in the literature of normal single limb lab rabbit kinematics. However, I do believe the authors could provide some further elaboration without collecting additional data.

Validity of the findings

The findings are valid, but require some additional elaboration and could benefit from additional analyses. Data were provided, but were not easy to decipher. An additional key or additional elaboration on column titles would be helpful.

The discussion requires more work, particularly in contextualizing the results with human biomechanics (or eliminating this comparison altogether).

Please also address issues with Figure 5. I might suggest that this figure be removed altogether (see my comments about using relative joint position without scaling and context).

The conclusions are appropriate given the results. However, the results and interpretation need some work before this manuscript is suitable for publication.

Additional comments

General comments:

The authors attempt to make links between human and rabbit mechanics, presumably for future work using rabbit models to clinical studies. However, these comparisons are very cursory and require a lot more discussion if this is going to be included. I also suggest the authors include a more significant summary of human biomechanics in the introduction to contextualize their discussion section. My suggestion is to consider dropping the human locomotor comparisons and only discuss their results in the context of other rabbit or half-bounding locomotor studies.

I also think that the authors need to either eliminate on human comparisons or remove them altogether. There are some factual errors and some glossing over comparisons that need to be addressed (e.g., rabbits don’t heel strike).

Finally, I would suggest the authors consider additional analyses that don’t require additional data collection. Why not look at stride length? Time of contact? Vertical force impulse? Time of contact and force impulse in particular would be useful for characterizing differences in limb loading when testing clinical interventions in future work. The authors also don’t quantify velocity, acceleration, etc.? Was this collected with steady state locomotion? Was there variation in joint angle that correlated with velocity? Were there differences between the animals? Some basic statistics may be warranted here. These analyses would make this a stronger manuscript with broader applicability.

Besancon et al. 2003 (Vet Comp Orthop Tramatol) and Lascelles et al. 2007 (Vet Record) provide examples of how others have used a pressure mat to characterize normal forces in an animal model. Additionally, previous work using other animal models such as rats have used swing and stance phase duration to characterize gait kinematics pre- and post-injury/intervention (e.g., Iwata et al. 2010 [Muscle Nerve].

Notes for future work:

I also recommend a different camera for future work comparing surgical interventions given that frame rates above 200 fps are required to detect compensatory gait patterns in rodents (Lakes and Allen 2016 [Osteoarthritis and Cartilage]). This number may be lower in rabbits, but my guess is that 60 fps may not be sufficient. Additionally, I would recommend including hip joint markers for a more complete picture of total limb kinematics.

Specific comments:

[Lines 93-95] Is it standard to only compute appropriate sample sizes in clinical trials? IACUC regulations mandate reductions in animal numbers through a variety of means, including power analyses. Is there a separate justification for why you chose n=6?
[Line 131] Most studies seem to calculate the MTP angle using the opposite direction. It’s not a big deal since that’s the complimentary angle, but just to note for the future.
[Lines 132-134] The ground is defined as the horizontal plane that runs through the MTP joint, but is this just at mid-stance? Please provide some more detail.
[Lines 138] I’m unclear what you mean by “knee joint center position”. Please clarify.

[Line 158-159] While I appreciate that the authors are making the case for using rabbit models to orthopedics, I’m not sure that a similar foot strike pattern is necessary. Many other animals exhibit similar ankle angles at heels strike, with similar loading patterns. However, I would argue that half bounding kinematics may be more relevant to human running than in other typical animal models like rodents.
[170-171] Characterizing the change in distance of the knee joint in the vertical and anterior-posterior dimensions strike me as an unusual kinematic analysis. In particular, it’s not size-scaled and not standard so it’s difficult to compare it to other work in the literature. I’m also not sure what you mean by “knee joint center position”. Why not just use joint angle over stance phase? I suggest removing the segment position results.
[175-176] It’s not particularly surprising that vGRF is correlated with contact area. The more body weight that is supported, the higher the vGRF. In fact, it should. Please add some context for this result. Is it a check on data quality?
[189-190] What do you mean by rabbits “load their heel first”? While plantigrade, rabbits don’t heel strike. Contact is on their metatarsals and toes. The heel is at the ankle joint.
[191-192] What do you mean by “foot angle during stance is more negative during human walking”? Negative compared to what? Please use refer to angles being smaller or larger. There are also other significant differences in locomotor mechanics than just joint angle (e.g., striding bipedal gait, limb-mass mechanics, etc.). Please elaborate if you are going to try to relate your results with human mechanics.
[202-207] Please elaborate on what you mean by “constant ground-paw pressure may be a locomotor strategy employed by the rabbit’s sensorimotor system”. Additionally, what do you mean by constant paw pressure? Pressure wasn’t constant as exhibited by the %BW force trace (e.g., changes in vGRF).

[208-215] I appreciate that injured animals may have biomechanical compensation, but how can you have different limb segment positions if joint angles are the same? Chang et al. found that limb length recovery tracked kinematic recovery. If animals are loading one limb less than the other, this probably means increased joint flexion, which means decreased joint center height (i.e., compensations should be correlated with joint angle). Additionally, as commented above, reporting 2D distances of joint center is difficult to interpret, not scaled, and not standard.
[217] I don’t agree that your system is ‘marker-less’. You used markers, just not reflective ones. Both reflective and ‘sharpie’ marks are still subject to skin movement over joint centers. There are marker-less systems for humans (e.g., Theia3D) which use multiple landmarks to locate joint centers.
[236] Though there are advantages to force plates, even with these trade-offs. One in particular for this study would be more precise center of pressure and impulse data.
[255] Again, I suggest removing comparison to humans unless a more thorough comparison is made. I don’t think it detracts from your study to keep contextualize this within rabbits or other animal models.

Figures:
[Fig 5] I find this figure really difficult to interpret. There is a grey shaded region showing the SD around the mean, but it’s really difficult to follow and interpret. Also, why are there no confidence bands around the ankle and MTP joints?

Reviewer 2 ·

Basic reporting

This manuscript clearly describes a well-designed study to characterize certain aspects of rabbit gait. It is well organized, includes appropriate references to the literature and provides generally clear figures to document methods and findings. The study is well motivated and discussion includes consideration of limitations.

The inclusion of raw data is helpful for further investigation of the study's results, however it would be helpful if the data workbook included more detailed headings, with explanation of some of the extraneous data provides at the bottom of columns.

For example, in the data set, two sets of "pressure" data are provided. Is one the raw data from the sensor and the other the calculated data?

It would be helpful if units were provided for each type of data.

Experimental design

The study's experimental design is well-defined and within the scope of the journal. The results are primarily of use as a "baseline" dataset for comparison for future studies, but that is quite valuable to the research community. It also describes methods for others to consider, with sufficient detail (in general) for replication.

One area that warrants further description is the selection of the five trials used for analysis. It seems clear from the dataset that multiple trials were needed to obtain the 5 used in the study for each animal/side. How were the final trials selected? It is well known that speed of gait influences human gait, but that doesn't appear to have been controlled in this study.

Further details on the walkway used and its sensitivity would be helpful. The authors state that they calibrated with a 3 kg weight, but not how large that was and thus what pressures were measured during calibration. Based on the information available about that walkway, it would seem that typical pressures were quite low relative the the range of the sensors.

Validity of the findings

The results of the study seem sound in general, although my calculations with the provided data do not seem to identify the same mean maximum ground reaction force. Perhaps I'm not understanding how the data are pooled between rabbits or by side? Clarification of the statistical methods may be needed.

Of some concern to this reviewer is the wide variability in the duration of the stance phase. Some trials are much slower than others, and those generally have lower ground reaction forces, as would be consistent with studies of human gait. The authors note this skewed distribution of the stance duration, but not the potential consequence on the final results. At a minimum, this should be discussed as a potential limitation. The patterns of the gait cycle seem different for these long duration trials, while in contrast, those that are very short have much higher peak forces. It isn't clear which of the two are more "typical" of rabbit gait that might be anticipated during a future experimental study. It may be appropriate to consider if any of the trials should be considered as "outliers". There does seem to be a modest correlation between the stance duration and the peak force.
It appears that in the figures, the data are converted to a %stance so that they can be depicted on a common graph. The variable patterns seen for trials at different speeds are likely contributing a great deal to the shaded region, but it almost seems like you are trying to show both walking and running on one plot.

Another area of concern is the potential for skin movement as an artifact for calculation of joint kinematics. This is a reasonable approach that is indeed more practical than other methods like x-ray or pins, however the skin motion could be significant. Again, the authors clearly state this potential limitation, but not the expected magnitude of the impact on the accuracy of the reported results. Would it be possible to calculate the lengths of the limb segments over the gait cycle and report on the degree of variability, which could reflect inaccuracies in identification of the joint centers.

Additional comments

It might be helpful to have scale bars in figure 3, both for the dimensions of the box and for the magnitude of the pressures measured.

---

## Round 0.2 · Minor Revisions

Dear Dr Crouch,

Thank for your thorough work responding to the reviewers’ comments on the first draft of your manuscript. As you’ll see, both reviewers have overall positive comments on the revision, with Reviewer 1 suggesting a few more minor revisions and Reviewer 2 recommending Acceptance. I agree with Reviewer 1 that there are few places where the manuscript could use some more clarification, as noted in the reviewer’s comments. I do NOT feel that you need to separately quantify speed from your dataset and reanalyze the results. The correlations with stance duration sufficiently speak to possible speed effects in the dataset. However, I would recommend adding an additional statement to the “limitations” paragraph at the end of the Discussion (right before Conclusions), suggesting that future studies in this area directly measure locomotor speed.

I’ll be happy to look over this last set of revisions myself. I do not think the manuscript will need an additional round of review. I look forward to seeing your revised manuscript.

Jesse W. Young
Department of Anatomy and Neurobiology
Northeast Ohio Medical University

Reviewer 1 ·

Basic reporting

The authors use clear, professional language, and have written an appropriate article in style and format, that is self-contained with relevant results and hypotheses.

Experimental design

This experiment is appropriate for PeerJ in scope and aims and fills a gap in the literature of normal single limb lab rabbit kinematics. This revised manuscript addresses reviewer concerns and is now sufficient for publication.

Validity of the findings

The findings are valid and the changes made to this manuscript make the interpretation of findings and discussion appropriate.

Additional comments

My apologies for not noting that you did indeed calculate time of contact (in this case, stance duration). I think in this case stride length is not nearly important as variables which measure loading, so this isn’t necessary.

Additionally, I would like to note that I do think that some form of velocity is an important variable to report. I would argue that with sufficient frame rates, using a marker (like the hip) can give reasonable measures of velocity and acceleration. Even if the distance/time measure is only when the rabbit is in view of the camera. In particular, it can give important information like whether rabbit trials were in steady state or accelerating. However, stance duration is certainly going to be correlated with speed.

Specific comments:

[Pg 3, Line 57] “Clinical” is spelled incorrectly

[Pg 3, Line 68] A minor quibble, but the phrasing in this paragraph initially sounds like you are reporting on your own data in the introduction. I think it would be helpful to say something like “In Gushue et al. (2005), a net vertical GRF curve showed XYZ...However these data only come from a sample of n=1, which makes it unclear if this is a typical gait pattern”, or something along those lines. This paragraph could also benefit from another sentence or two as it is a little terse. Can you elaborate just a little bit on the contrast you setup between rabbits having a “plantigrade type” foot morphology and a digitigrade gait pattern? Because it immediately jumps to a description of results from a paper that wasn’t cited in the previous sentence. I think this will help setup your study to make it clear what is known, unknown, and how your study addresses a gap in the literature. I think this adjustment will make for a stronger introduction and setup for the rest of the paper

[Pg 9, Line 216-218] In what ways were rabbit kinematics similar to rats during walking?

Reviewer 2 ·

Basic reporting

The manuscript is clearly written and has been improved by modifications in response to both reviewers. The provided dataset is much more clearly presented.

Experimental design

The methods and experimental design are clearly articulated and sufficient detail is provided to replicate the experiment.

Validity of the findings

The findings appear to be valid based on the limited experiment conducted. There are surely limitations due to skin motion, but these are clearly acknowledged and are not likely to change the general conclusions.

---

## Round 0.3 · accepted · Accept

Thank you for your quick work in responding to all remaining comments.